# Plasma-Free Strategy for Cardiac Surgery with Cardiopulmonary Bypass in Infants < 10 kg: A Retrospective, Propensity-Matched Study

**DOI:** 10.3390/jcm12123907

**Published:** 2023-06-07

**Authors:** Marco Ranucci, Umberto Di Dedda, Giuseppe Isgrò, Alessandro Giamberti, Mauro Cotza, Noemi Cornara, Ekaterina Baryshnikova

**Affiliations:** 1Department of Cardiovascular Anesthesia and Intensive Care, IRCCS Policlinico San Donato, San Donato Milanese, 20097 Milan, Italy; umberto.didedda@grupposandonato.it (U.D.D.); giuseppe.isgro@grupposandonato.it (G.I.); mauro.cotza@grupposandonato.it (M.C.); noemi.cornara@gmail.com (N.C.); ekaterina.baryshnikova@grupposandonato.it (E.B.); 2Department of Congenital Heart Surgery, IRCCS Policlinico San Donato, San Donato Milanese, 20097 Milan, Italy; alessandro.giamberti@grupposandonato.it

**Keywords:** fresh frozen plasma, pediatric cardiac surgery, plasma-free strategy, postoperative bleeding, transfusions

## Abstract

Background: Infants < 10 kg undergoing cardiac surgery with cardiopulmonary bypass (CPB) may receive either fresh frozen plasma (FFP) or other solutions in the CPB priming volume. The existing comparative studies are controversial. No study addressed the possibility of total avoidance of FFP throughout the whole perioperative course in this patient population. This retrospective, non-inferiority, propensity-matched study investigates an FFP-free strategy compared to an FFP-based strategy. Methods: Among patients <10 kg with available viscoelastic measurements, 18 patients who received a total FFP-free strategy were compared to 27 patients (1:1.5 propensity matching) receiving an FFP-based strategy. The primary endpoint was chest drain blood loss in the first 24 postoperative hours. The level of non-inferiority was settled at a difference of 5 mL/kg. Results: The 24-h chest drain blood loss difference between groups was −7.7 mL (95% confidence interval −20.8 to 5.3) in favor of the FFP-based group, and the non-inferiority hypothesis was rejected. The main difference in coagulation profile was a lower level of fibrinogen concentration and FIBTEM maximum clot firmness in the FFP-free group immediately after protamine, at the admission in the ICU and for 48 postoperative hours. No differences in transfusion of red blood cells or platelet concentrate were observed; patients in the FFP-free group did not receive FFP but required a larger dose of fibrinogen concentrate and prothrombin complex concentrate. Conclusions: An FFP-free strategy in infants < 10 kg operated with CPB is technically feasible but results in an early post-CPB coagulopathy that was not completely compensated with our bleeding management protocol.

## 1. Introduction

Small infants undergoing cardiac surgery with cardiopulmonary bypass (CPB) usually require allogeneic red blood cells (RBC) to prime the CPB circuit and oxygenator in order to maintain an acceptable hematocrit (HCT) during the procedure. To maintain a physiological oncotic pressure, RBC in the priming volume is diluted with fresh frozen plasma (FFP), 5% albumin, or gelatins.

Different studies compared the FFP-based priming with either albumin [1,2,3,4], gelatins [5,6] or crystalloids [7]. The results of these studies are conflicting. The majority concludes that FFP can be safely substituted with other solutions without increased bleeding or transfusional needs [2,3,5,6,7], but others [1,4] found larger postoperative bleeding or more allogeneic blood products transfusions when an FFP-free priming was used.

Basically, avoidance of FFP in the priming solution leads to a dilution of fibrinogen and soluble coagulation factors immediately after CPB [4,8]. This can be compensated with post-CPB administration of FFP. Actually, all the studies comparing FFP-based priming vs. no FFP in the priming allowed the use of FFP after protamine administration and during the intensive care unit (ICU) stay [1,2,3,4,5,6,7]. However, both fibrinogen and soluble coagulation factors can be replaced using fibrinogen concentrate and prothrombin complex concentrate (PCC), avoiding the need for FFP at any perioperative time. This FFP-free strategy is well-established in the adult population [9]. Conversely, it is not in newborns and small infants: the recently released guidelines of the Network for the Advancement of Patient Blood Management, Haemostasis and Thrombosis (NATA) [10] recommend the addition of FFP to the CPB prime in neonates (<30 days) at a level 2C.

Based on the theoretical feasibility of FFP substitution with fibrinogen concentrate and PCC and our preliminary experience [8], where low postoperative levels of fibrinogen and prothrombin activity were identified as the main predictors of severe bleeding, we started employing an FFP-free strategy in infants < 10 kg in 2019. The purpose of the present retrospective, propensity-matched study is to test the experimental hypothesis that a total FFP-free strategy is not inferior to an FFP-based strategy (in the priming volume and throughout the perioperative period and the ICU stay). The primary endpoint of the study was the chest drain blood loss difference in the first 24 postoperative hours.

## 2. Materials and Methods

This is a retrospective, propensity-matched, non-inferiority study including patients operated at our institution between 2015 and 2021. The study was approved by the local Ethics committee of the San Raffaele Hospital (11 May 2022, protocol number 55/INT/2022). Given the retrospective nature of the study, written informed consent was retrieved whenever feasible; all the patients, however, provided written informed consent for the use of their data for scientific research in an anonymous form. The study was partially funded by the Italian Ministry of Health, within the funding of the Clinical Research Hospitals network, which includes our Institution.

The primary endpoint of the study was the difference in chest drain blood loss in the first 24 postoperative hours. Secondary endpoints were the incidence of surgical revision due to bleeding, the amount of allogeneic blood products transfused, the use of fibrinogen and prothrombin complex concentrate, and the hemostatic profile.

### 2.1. Patient Population and Propensity Matching

The FFP-free strategy in infants < 10 kg was tested starting in 2019. This strategy was not a routine practice and was applied to a limited patient population. However, no pre-specified selection criteria were applied, and the FFP strategy was chosen after an agreement between the cardiac surgeon, the anesthesiologist, and the perfusionist. Figure 1 shows the flowchart leading to the final sample size. From January 2015 through May 2021, 964 infants < 10 kg were operated with CPB at our institution, with an FFP-based strategy (FFP in the priming volume and after CPB, from January 2015 through May 2021) or an FFP-free strategy (from January 2019 only).

The first selection criterion was the availability of viscoelastic tests (VET) throughout the first 48 postoperative hours, namely (a) immediately after protamine administration, (b) after 24 h from surgery, and (c) after 48 h from surgery. All the FFP-free patients received VET at these three points in time as a routine measure for this strategy; conversely, in the FFP-based group, VET was not routinely applied after CPB (unless in the presence of microvascular bleeding), nor in the first 48 postoperative hours (unless in the presence of clinically relevant chest drain blood loss). Therefore, to avoid a selection bias (i.e., the inclusion of bleeding patients only) in the FFP-based group, we included patients from our previous studies [4,8], where these measures were available regardless of the bleeding pattern. This led to a patient population of 81 potentially eligible patients.

A second selection criterion was the exclusion of patients requiring postoperative Extra Corporeal Membrane Oxygenation (ECMO). This led to a patient population of 75 patients (18 in the FFP-free group and 57 in the FFP-based group) admitted to the further steps of propensity-matching (Figure 1).

The propensity matching process followed the current state of the art [11,12]. Basically, we performed a logistic regression model fitted with treatment status (FFP-free or FFP-based strategy) as the outcome; explanatory variables suspected of being confounders had to fulfill these categories: (i) occur temporarily before the treatment or the outcome measure; (ii) are associated with the treatment versus control decision; and (iii) are associated with the outcome at a level of absolute standardized difference (ASMD) > 0.10.

The characteristics of the study population (FFP-free group, pre-matching FFP-based group, and post-matching FFP-based group) are shown in Table 1. A 1.5:1 matching process was performed, extracting cases from the FFP-based group with the same propensity score as the cases in the treated group.

The correct matching was checked through an analysis of the absolute standardized mean difference after matching (Table 1). According to the current standards, an ASMD < 0.15 is considered a very small effect size, and between 0.15 and 0.20 is a small effect size [13]. After propensity matching, the between-groups ASMD never exceeded the value of ±0.20, with 9 items carrying a very small effect size and 3 carrying a small effect size.

### 2.2. Anesthesia, CPB, Surgery, and ICU Management

Every subject received our standard surgical treatment and CPB technique. A total intraoperative dose of 30 mg/kg of tranexamic acid was given to every patient. CPB was established after a loading dose of 300 IU/kg of unfractionated heparin plus additional doses (80 IU/kg) to reach and maintain a target-activated clotting time of 450 s or longer. The target patient temperature was chosen based on the type of surgical procedure and cardioplegia protocol. Ultrafiltration was a standard of care: conventional or modified ultrafiltration was applied, respectively, during and after CPB, according to the surgeon’s preferences. The CPB circuit included a hollow fiber oxygenator (Sorin KIDS D100 or D101, Livanova, Mirandola, Italy), a roller head pump (Sorin S5 HLM, Livanova, Mirandola, Italy) or a centrifugal pump (Bio-Medicus, Medtronic, Minneapolis, MN, USA).

### 2.3. Interventions

CPB priming of patients in the FFP-free group was formulated with albumin 5% plus RBC. Patients in the FFP-based group received a priming solution with FFP plus RBC. The solution was titrated to reach an “on pump” hematocrit of 30%. The number of RBCs used in the priming solution varied according to the patient’s baseline HCT, weight, and priming volume. The “clear prime volume” (albumin 5% or FFP) was obtained as the difference between the circuit priming volume and the calculated amount of RBC.

Every volume addition needed during CPB was made by giving albumin 5% (FFP-free group), or FFP, and/or RBC according to our HCT target.

### 2.4. Data Collection and Measurements

Patient characteristics, preoperative data, type of surgery, CPB parameters and outcome data (morbidity and mortality, mechanical ventilation time, ICU stay, and postoperative hospital stay) were extracted from our institutional database. Data relative to the primary and secondary endpoints were collected as follows: (i) chest drain blood loss at 12, 24, and 48 postoperative hours, and expressed in terms of mL kg^−1^ and cumulative value; (ii) transfusion of FFP, platelet concentrate, and RBC were expressed in terms of mL kg^−1^, and were analyzed separately for priming volume, intraoperative, from arrival in ICU to the following 24 h, from 24 h to 48 h, and cumulative; (iii) use of fibrinogen concentrate and PCC was expressed as mL kg^−1^ and cumulative amount in the OR and ICU.

Standard coagulation tests (activated partial thromboplastin time [aPTT, seconds], prothrombin time [PT, %] activity, platelet count, fibrinogen concentration) were performed preoperatively (excepted fibrinogen), at the arrival in the ICU, after 24 and 48 h from surgery. VET (ROTEM, TEM International, Munich, Germany) were performed after protamine administration and after 24 and 48 h from surgery and included EXTEM, INTEM, HEPTEM, and FIBTEM tests.

### 2.5. Bleeding Treatment Protocol

Post-CPB bleeding was treated based on the VET results in the operating room (OR) and on VET results plus standard coagulation tests in the ICU. Our protocol was based on a vertical algorithm that was applied only in the presence of microvascular bleeding in the OR or excessive chest drain blood loss in the ICU. The protocol was based on the following step-by-step interventions:Correction of residual heparin, if the CT at INTEM exceeded by 20%, the CT at HEPTEMFibrinogen concentrate (Haemocomplettan or Riastap, CSL Behring, Marburg, Germany) 30 mg kg^−1^ if FIBTEM MCF < 8 mm or Clauss fibrinogen < 150 mg dL^−1^ in the FFP-free group, and/or FFP in the FFP-based group.Platelet concentrates if EXTEM MCF < 35 mm and FIBTEM MCF ≥ 8 mm or platelet count < 100,000 cells/µL. In case of persisting bleeding with an EXTEM MCF < 35 mm, platelet concentrates could be repeated regardless of FIBTEM MCF values.4-factors PCC (Confidex, CSL Behring, Marburg, Germany or Pronativ, Octapharma, Lachen, Switzerland) 20 IU kg^−1^ if EXTEM CT > 100 (after correction of fibrinogen and platelet values) or prothrombin activity < 50% in the FFP-free group, and/or FFP in the FFP-based group.Additional tranexamic acid (30 mg kg^−1^) in the presence of VET signs of hyperfibrinolysis.Surgical re-exploration if bleeding persisted after correction of the above factors.

RBCs were administered in order to maintain an HCT value between 30% and 35%, depending on the nature of the cardiac defect and surgery, the hemodynamic state, and the presence of active bleeding.

### 2.6. Sample Size and Statistics

The sample size was based on a non-inferiority hypothesis of the FFP strategy in terms of 24-h chest drain blood loss. The reference value for the FFP-based group was settled at 24 mL kg^−1^ with a standard deviation (SD) of 13 mL kg^−1^ based on our previous study [4]. The mean between-group differences were settled at 6 mL kg^−1,^ and the clinically relevant difference was settled at 5 mL kg^−1^. With an FFP-based to FFP-free ratio of 1.5:1, an alpha value of 0.05 and a power of 80%, the sample size for the non-inferiority hypothesis is 38 patients (15 FFP-free and 23 FFP-based). Having 18 FFP-free patients, we increased the sample size to 45 patients (18 FFP-free and 27 FFP-based).

Data are expressed as mean (standard deviation) or median (interquartile range) depending on the normality of distribution and as numbers (%). Comparisons between groups included the measure of mean differences with 95% confidence intervals (CI); differences in continuous variables were investigated with a Student’s *t*-test or non-parametric tests, when appropriate; differences in dichotomous variables were tested with a Pearson’s chi-square. All the statistical analyses were performed with computerized packages (SPSS 20.0, IBM, Chicago, IL, GraphPad, GraphPad Software, Inc, San Diego, CA, and MedCalc, MedCalc Software, Ostend, Belgium). A two-tailed *p* value < 0.05 was considered significant for all the statistical tests.

## 3. Results

Overall, the most frequent surgery was ventricular septal defect repair with or without associated procedures (14 patients, 31.1%), followed by cavopulmonaryconnection in various kinds of single ventricle pathology (7 patients, 15.5%), aortic coarctation with aortic arch involvement (6 patients, 13.3%), and truncus arteriosus repair (4 patients, 8.9%).

Table 2 reports the preoperative and postoperative coagulation profiles for the standard and viscoelastic coagulation tests. The preoperative profile was not significantly different between groups. Immediately after protamine administration, the VET showed a significantly (*p* = 0.001) prolonged CT, a significantly (*p* = 0.001) lower EXTEM MCF, and a significantly (*p* = 0.001) lower FIBTEM MCF. At the arrival in the ICU, the standard coagulation tests showed a significantly (*p* = 0.001) longer aPTT, a significantly (*p* = 0.001) lower PT activity, and a significantly (*p* = 0.002) lower fibrinogen concentration, with no differences in platelet count. At the subsequent standard coagulation tests and VET analyses, the PT activity remained significantly (*p* = 0.048) lower in the FFP-free group at 24 h, and the fibrinogen concentration remained significantly lower at 24 and 48 h; the EXTEM MCF remained significantly (*p* = 0.041) lower at 24 and 48 h, and the FIBTEM MCF remained significantly (*p* = 0.001) lower at 24 and 48 h.

Bleeding and transfusions (RBC, FFP, platelet concentrate) and blood derivates use shown in Table 3. The primary endpoint (bleeding at 24 h) was not significantly different between groups, as well as cumulative bleeding at 48 h. However, (Figure 2) the non-inferiority hypothesis for the FFP-free strategy was not confirmed since the 95% confidence interval of the bleeding difference at 48 h exceeded the level of 5 mL kg^−1^, which was considered as the clinically relevant difference in our study. The difference in bleeding, albeit not significant, was in favor of the FFP-based group at 12, 24, and 48 h.

RBC transfusions did not differ between groups, as well as platelet concentrate transfusions. Because of the study protocol, no patient in the FFP-free group received FFP, which was used in a significantly larger dose in the priming volume and in OR in the FFP-based group. Finally, the use of fibrinogen concentrate was significantly (*p* = 0.001) larger in the FFP-free group, as well as the use of PCC (*p* = 0.020).

The HCT at the arrival in the ICU was significantly (*p* = 0.013) higher in the FFP-free group (41.5%) vs. the FFP-based group (38.1%), with a mean difference of 3.4% and a 95% CI 0.8 to 5.9%. At 24 h, there was no significant difference (FFP-free: 38.9%, FFP-based: 37.6%, mean difference 1.3%, 95% CI −1.5 to 4.2%, *p* = 0.355), and at 48 h, the HCT was again significantly (*p* = 0.003) higher in the FFP-free group (39.4%) vs. the FFP-based group (35.8%), with a mean difference of 3.5%, 95% CI 1.2 to 5.9%.

The overall outcome of the two groups did not significantly differ, with the exception of a higher peak serum creatinine level in the FFP-free group (Table 4).

## 4. Discussion

Based on our primary endpoint and on the requisites of a non-inferiority study [14], the non-inferiority hypothesis of an FFP-free strategy vs. a liberal use of FFP in the CPB priming and after surgery in small infants was not confirmed. Actually, the correct conclusion is that the FFP-free strategy is “neither inferior or non-inferior” [14] to the FFP-based strategy, but equivalence or superiority of the FFP-based strategy cannot be claimed as well.

Among the secondary endpoints, no differences were detected for red blood cell or platelet concentrate transfusions. As a consequence of the study design, there was a larger use of FFP in the FFP-based group and of fibrinogen concentrate and PCC in the FFP-free group. No differences were found with respect to the time of mechanical ventilation, ICU stay, postoperative hospital stay, morbidity and mortality, unless for a higher postoperative peak creatinine value in the FFP-free group. This last finding could be related to a higher kidney protein (fibrinogen) load in the FFP-free group.

Of notice, and despite a trend towards larger bleeding and the same amount of RBC transfused, the FFP-free group had higher HCT values at the arrival in the ICU and after 48 postoperative hours. This is probably a consequence of the haemodilutional effect of the relatively large doses of FFP administered in the FFP-based group in the OR and the ICU.

The hemostatic profile offers a number of insights for the interpretation of clinical data. Immediately after CPB and at the arrival in the ICU, the FFP-free group showed clear signs of dilution and consumption of fibrinogen and soluble coagulation factors, with VET signs of prolonged EXTEM CT and reduced clot firmness (EXTEM MCF) mainly due to a poor FIBTEM MCF. Standard coagulation tests showed a prolonged aPTT, decreased PT activity, and very low fibrinogen levels. This was already observed in our preliminary studies, where no FFP was used in the priming volume [4,8] and by other authors who compared an FFP-free priming volume with an FFP-based priming volume. McCall and colleagues [1] found a significantly lower level of fibrinogen in the group without FFP in the priming that, however, was normalized at the ICU admission. Oliver and co-workers [2] used thromboelastography (TEG), together with standard coagulation tests and found that after CPB, the group with 5% albumin in the priming volume had significantly longer prothrombin time and lower fibrinogen levels. No differences in TEG parameters were found at the end of CPB. However, the fibrinogen contribution to clot firmness was not addressed. At the arrival in the ICU, no differences in standard coagulation tests were observed. Lee and colleagues [3] compared albumin priming to FFP priming using a ROTEM analysis. They found that immediately after protamine administration, there was a significantly lower clot firmness at INTEM, EXTEM and FIBTEM MCF, with no differences in reaction times. Again, these differences were no more statistically significant after 24 h in the ICU. Similar results were obtained by Miao [5], who tested the hemostatic profile with TEG: the FFP-free priming group showed significantly lower values of fibrinogen contribution to clot firmness. Finally, a randomized controlled trial [7] found that immediately after CPB, the FFP-free priming group had a reduced MCF at EXTEM and FIBTEM with a prolonged clot formation time at EXTEM. All the parameters were not significantly different from the FFP-based priming group at the arrival in the ICU.

These studies report results that are only partially comparable to ours. The first difference is that some authors included patients older and/or of larger weight than our patient population, ranging from 1 month to 16 years [3], 6 months to 3 years [5], 7 to 15 kg [7], with two studies only including a patient population < 10 kg [1,2]. The second and most important difference is that all the other studies admitted the use of FFP and cryoprecipitate after protamine administration and in the ICU, whereas this was totally avoided in our study. As a consequence, only data immediately after protamine administration can be compared with our results, and to this respect, there is a general consensus that avoiding FFP in the priming volume generates the pattern of a dilution/consumption coagulopathy mainly affecting the fibrinogen contribution to clot firmness.

Conversely, all the other studies found no differences in the hemostatic pattern since the arrival in the ICU, while in our study, the differences remained significant at the arrival in the ICU, and for some items (fibrinogen concentration and contribution to clot strength) even after 24–48 h. This demonstrates that our strategy based on compensation of the dilution/consumption coagulopathy was less efficient than a strategy based on FFP and/or cryoprecipitate. It may be argued that our strategy was inadequate to correctly restore the fibrinogen and soluble coagulation factors level due to too restrictive trigger values and/or insufficient doses. In particular, it is possible that larger fibrinogen concentrate doses may be needed to correctly restore fibrinogen levels. In a recent study [15] on a patient population similar to ours, Siemens and co-workers found similar levels of FIBTEM at the end of CPB (5.3 mm) and corrected them (up to 13 mm) with a much larger fibrinogen concentrate dose than what we used (median dose 114 mg/kg vs. 30 mg/kg). As a matter of fact, trigger values for prompting fibrinogen and soluble coagulation factors replacement are far from being established in small infants. The trigger values used in adults are probably inadequate, especially considering that the physiological level of both coagulation factors and natural anticoagulants are lower in small infants (developmental hemostasis) [16].

The main limitation of our study is its retrospective nature. Additional limitations are the lack of information on preoperative fibrinogen levels and data on surgical blood loss before sternal closure.

In conclusion, our study could not demonstrate a non-inferiority of a total FFP-free strategy in infants < 10 kg receiving cardiac surgery with CPB. However, our data suggest that the strategy is technically feasible without major complications and that a trend toward a larger chest drains blood loss did not result in a larger use of RBC, probably due to a limitation of FFP-related hemodilution. The bleeding management algorithm was probably undersized to correctly restore the dilution/consumption of soluble coagulation factors and fibrinogen. Our conclusion, in agreement with the existing guidelines [10], is that FFP is preferable to 5% albumin in the priming volume to avoid the onset of severe dilution/consumption coagulopathy. Conversely, after CPB, correction of low fibrinogen levels is better with fibrinogen concentrate than with FFP, whose large doses induce a dilutional effect. Complete elimination of FFP is certainly a good target in the setting of patient blood management, but it is certainly a difficult, resources demanding strategy, burdened by the higher costs of the large doses of fibrinogen and PCC to correct severe post-CPB coagulopathy. Additional, larger trials addressing the limitations of the present study are certainly needed to further elucidate the options for hemostasis optimization in the pediatric population.

## Figures and Tables

**Figure 1 jcm-12-03907-f001:**
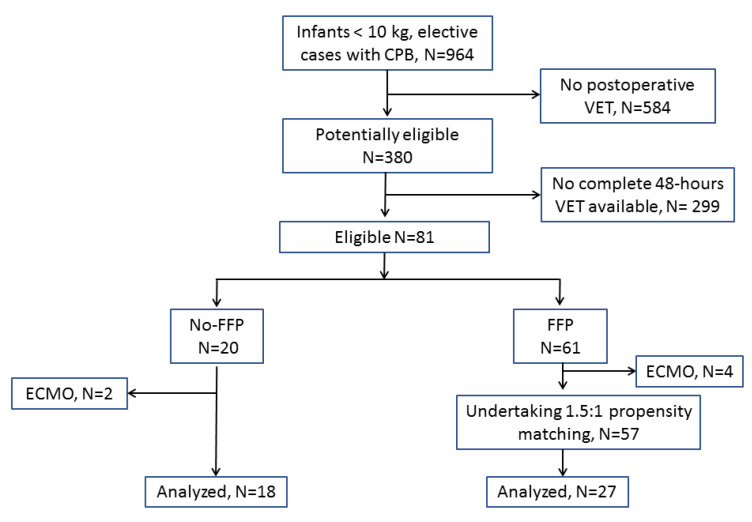
Flowchart leading to the final sample size. CPB, cardiopulmonary bypass; ECMO, extracorporeal membrane oxygenation; FFP, fresh frozen plasma; VET, viscoelastic tests.

**Figure 2 jcm-12-03907-f002:**
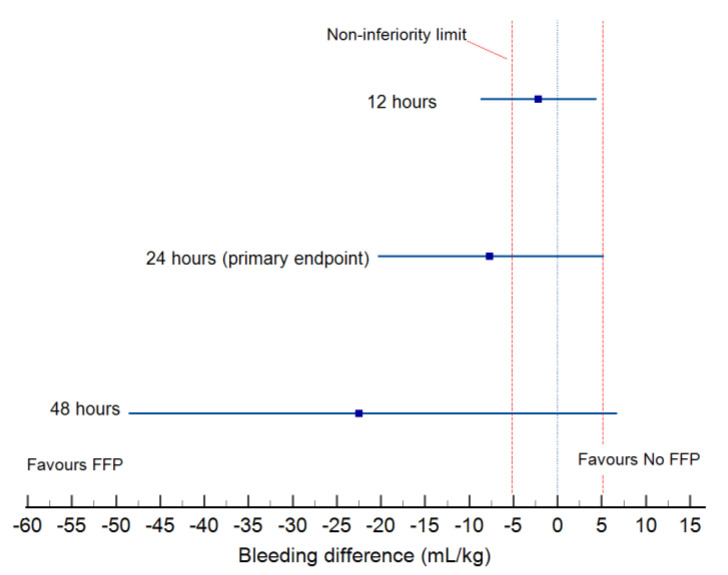
Bleeding difference between groups at 12, 24 and 48 h after surgery.

**Table 1 jcm-12-03907-t001:** Pre- and intra-operative variables before and after matching.

Variable	FFP-Free (*n* = 18)	FFP-Based before Matching (*n* = 53)	ASMD Pre-Match	FFP-Based after Matching (*n* = 27)	ASMD Post-Match
Age (months)	7.00 (7.0)	7.02 (7.3)	0.00	6.3 (4.6)	0.12
Weight (kg)	5.46 (2.01)	5.62 (2.14)	−0.08	5.6 (2.0)	−0.07
Preoperative hematocrit (%)	39.4 (6.3)	36.6 (7.5)	0.40	38.5 (8.6)	0.12
aPTT (sec)	33.2 (12.5)	32.7 (4.9)	0.05	32.4 (4.9)	0.08
PT (%)	88.3 (20.5)	90.3 (13.1)	−0.12	90.8 (13.7)	−0.14
Platelet count (×1000 cells/µL)	363 (113)	334 (122)	0.25	357 (136)	0.05
Cardiopulmonary bypass time (min)	91 (38)	108 (48)	−0.39	96 (45)	−0.12
Nadir hematocrit on CPB (%)	30.2 (3.2)	30 (2.8)	0.07	30.7 (3.0)	−0.16
Nadir temperature on CPB (°C)	31.1 (3.6)	30.9 (2.9)	0.06	31.6 (3.1)	−0.15
RACHS II	2.5 (1.5)	2.1 (0.91)	0.32	2.36 (1.05)	0.15
Redo surgery	4 (22.2)	4 (7.5)	0.42	5 (18.5)	0.09
Urgent surgery	5 (27.8)	8 (15.1)	0.31	7 (25.9)	0.04

Data are mean (standard deviation) or number (%). ASMD: absolute standardized mean difference (FFP-free vs. FFP-based groups). aPTT: activated partial thromboplastin time; CPB: cardiopulmonary bypass; FFP: fresh frozen plasma; PT: prothrombin activity; RACHS: Risk Stratification for Congenital Heart Surgery.

**Table 2 jcm-12-03907-t002:** Standard laboratory and viscoelastic tests main coagulation parameters.

Item		FFP-Free (*n* = 18)	FFP-Based (*n* = 27)	Mean Difference (95% CI)	*p*
aPTT (sec)					
	Baseline	33.2 (12.5)	32.7 (4.9)	−0.87 (−6.24 to 4.48)	0.743
	Arrival ICU	39.0 (7.5)	33.6 (4.1)	−5.5 (−9.0 to −2.2)	0.003
	24 h	37.8 (8.3)	36.5 (7.5)	−1.32 (−6.14 to 3.48)	0.581
	48 h	40.4 (7.5)	40.7 (11.3)	0.22 (−6.66 to 7.11)	0.948
Prothrombin activity (%)					
	Baseline	88.3 (20.5)	90.3 (13.1)	2.53 (−9.83 to 14.9)	0.679
	Arrival ICU	48.8 (11)	69.1 (10.5)	20.3 (12.8 to 27.8)	0.001
	24 h	55.3 (12.6)	62.3 (8)	7.54 (0.06 to 15)	0.048
	48 h	57.8 (9.4)	65.3 (12.9)	7.52 (−5.64 to 20.7)	0.250
Platelet count (×1000 cells/µL)					
	Baseline	363 (113)	357 (136)	−6.5 (−85 to 72)	0.868
	Arrival ICU	147 (48)	146 (74)	−1.7 (−41 to 38)	0.931
	24 h	153 (51)	146 (54)	−7.8 (−41 to 25)	0.635
	48 h	141 (50)	159 (71)	18.4 (−20 to 57)	0.345
Fibrinogen (mg dL^−1^)					
	Arrival ICU	156 (50)	202 (45)	46 (17 to 75)	0.002
	24 h	214 (71)	258 (55)	44 (6 to 82)	0.023
	48 h	271 (110)	393 (91)	123 (50 to 195)	0.002
EXTEM CT (sec)					
	Post-protamine	108 (27)	93 (39)	−14.8 (−36.5 to 6.98)	0.178
	24 h	88 (19)	78 (16)	−10.2 (−21 to 0.6)	0.064
	48 h	82 (20)	87 (22)	5.1 (−8.8 to 19)	0.462
HEPTEM CT (sec)					
	Post-protamine	312 (76)	272 (92)	−40.3 (−101 to 20.5)	0.186
	24 h	294 (107)	290 (138)	−4.1 (−91.5 to 83)	0.924
	48 h	270 (81)	247 (70)	−23.2 (−86 to 39.6)	0.454
EXTEM MCF (mm)					
	Post-protamine	44 (8)	52 (8)	8.7 (3.6 to 13.8)	0.001
	24 h	54 (8)	58 (6)	4.4 (0.20 to 8.7)	0.041
	48 h	55 (10)	60 (6)	5.5 (0.22 to 10.7)	0.041
FIBTEM MCF (mm)					
	Post-protamine	4.7 (2.1)	11 (6.3)	6.3 (3.2 to 9.4)	0.001
	24 h	10 (3.8)	15.7 (6.2)	5.7 (2.3 to 9)	0.001
	48 h	12.5 (6.5)	20.5 (5.9)	8.0 (3.9 to 12)	0.001

Data are mean (standard deviation). aPTT: activated partial thromboplastin time; CT: clotting time; FFP: fresh frozen plasma; ICU: intensive care unit; MCF: maximum clot firmness.

**Table 3 jcm-12-03907-t003:** Bleeding, transfusions, and blood components were used in the two groups.

Item	FFP-Free (*n* = 18)	FFP-Based (*n* = 27)	Mean Difference (95% CI)	*p*
Bleeding (cumulative, mL kg^−1^)				
12 h	16.7 (12)	14.5 (9.4)	−2.2 (−8.7 to 4.3)	0.501
24 h	31.9 (27.1)	24.2 (16.3)	−7.7 (−20.8 to 5.3)	0.239
48 h	58.5 (66)	37 (25.6)	−21.5 (−49.8 to 7.8)	0.133
Packed red cells transfusions (mL kg^−1^)				
Priming volume	22.9 (7.2)	23.1 (8.2)	2.4 (−4.7 to 4.9)	0.863
After protamine in the OR	11.5 (7.1)	11 (8.4)	2.4 (−5.4 to 4.3)	0.821
Postoperative 24 h	11.2 (9.3)	11.6 (9.5)	2.9 (−5.3 to 6.24)	0.875
Postoperative 48 h	4.3 (7.3)	2.3 (6.0)	−2 (−6 to 2)	0.312
Total	50 (19.2)	48 (23)	−2 (−15 to 11)	0.761
FFP transfusions (mL kg^−1^)				
Priming volume	0	29.4 (13.8)	29.4 (22.8 to 36)	0.001
After protamine in the OR	0	4.4 (5.1)	4.4 (2 to 6.9)	0.001
Postoperative 24 h	0	1.8 (5.9)	1.9 (−0.01 to 3.74)	0.051
Postoperative 48 h	0	0	0	N/A
Total	0	8.1 (18.3)	33.9 (26.3 to 41.4)	0.001
Platelet transfusions (mL kg^−1^)				
After protamine in the OR	1.14 (3.3)	1.28 (3.5)	0.15 (−1.95 to 2.24)	0.890
Postoperative 24 h	0	0.18 (1.0)	0.19 (−0.27 to 0.64)	0.420
Postoperative 48 h	0	0	0	N/A
Total	1.13 (3.3)	1.46 (3.8)	0.33 (−1.9 to 2.6)	0.766
Fibrinogen concentrate (mg kg^−1^)				
OR and ICU	23.4 (20)	4.3 (10.5)	−19.1 (−28.3 to −9.84)	0.001
PCC (IU kg^−1^)				
OR and ICU	5.6 (9.2)	0.74 (3.8)	−4.8 (−8.8 to −0.81)	0.020

Data are mean (standard deviation). FFP: fresh frozen plasma; ICU: intensive care unit; N/A: not applicable; OR: operating room; PCC: prothrombin complex concentrate.

**Table 4 jcm-12-03907-t004:** General outcome.

Item	FFP-Free (*n* = 18)	FFP-Based (*n* = 27)	*p*
Surgical revision due to bleeding	0 (0)	0 (0)	N/A
Inotropic drugs > 48 h	9 (50)	13 (52)	0.897
Bloodstream infection	3 (17)	1 (4)	0.190
Hospital mortality	1 (5.6)	0 (0)	0.419
Peak serum creatinine (mg dL^−1^)	0.59 (0.20)	0.46 (0.16)	0.020
Blood lactates arrival ICU (mmol L^−1^)	1.85 (1.30)	1.74 (0.67)	0.722
Mechanical ventilation (hours)	77 (84)	53 (55)	0.257
ICU stay (days)	4 (2–9)	4 (2–7)	0.674
Postoperative hospital stay (days)	15 (12–23)	14 (8–17)	0.236

Data are mean (standard deviation) or median (interquartile range) or number (%). FFP: fresh frozen plasma; ICU intensive care unit; N/A: not applicable.

## Data Availability

The original dataset supporting the findings of this study will be deposited in the public repository Zenodo after the publication of the work and accessible upon a reasonable request. The requests should be addressed to the corresponding author of the Manuscript.

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
