# Peer review of "Plasma-Free Strategy for Cardiac Surgery with Cardiopulmonary Bypass in Infants < 10 kg: A Retrospective, Propensity-Matched Study"

_jcm, 2023, doi:10.3390/jcm12123907_

Round 1

Reviewer 1 Report

Dear authors, 

please find my comments below.

Ranucci et al. present an retrospective propensity matched study comparing FFP with a FFP-free ECLS priming in infants < 10kg undergoing cardiac surgery. Primary endpoint was 24 chest tube drain. The authors describe a lower fibrinogen concentration in the FFP-free group, but did not observe clinically relevant outcome differences.

The present study is clear structured and clearly answers a clearly stated hypothesis after a well structured methodology. The data favors a FFP-free strategy, because its non-inferiority compared to the FFP-strategy. 

Thus, the FFP-free strategy can help to avoid a de fecto a priori transfusion of blood products with all potential side-effects (allergic reactions, infections, etc. ...)

With all its methodological limitations, this small study represents in a scarce literature environment a worthy contribution.

Author Response

We would like to thank the reviewer for his appreciation of the manuscript’s content.

Reviewer 2 Report

Dr. Ranucci and colleagues in their manuscript entitled “Plasma-Free Strategy For Cardiac Surgery With Cardiopulmonary Bypass In Infants < 10 Kg” studied the role of FFP-free priming for cardiopulmonary by-pass in infants undergoing cardiac operations.

This small propensity matched analysis showed that while avoiding FFP is possible, it results in more pronounced coagulopathy requiring infusions of fibrinogen and PCC. The manuscript is well written and touches an important topic. The conclusions are fair and supported by the data.

Major:

1.       As with many studies on bleeding in cardiac surgery the main issue is the choice of the endpoint. Please provide the rationale for choosing a numerical endpoint of lost blood volume instead of clinical endpoints, such as transfusions. Please provide rationale for choosing the blood loss in 24h instead of 12h, as used in the Universal Definition of Perioperative Bleeding in Cardiac Surgery for adults.

2.       The study involved small children unable to consent. Did the parents of studied children provide a written informed consent for study participation? Since the research involves minors, please outline the details of informed consent process.

3.       How many patients in each group were operated in deep hypothermic circulatory arrest and how many required open chest management? Can you provide some details on the operations performed?

4.       What was the c-statistics value for the final model.

5.       Which matching method was used?

Minor

1.       There is a redundant dot at the end of the abstract.

2.       Page 2, line 56: based on…? Line 59 – experimenting with?

Author Response

Dr. Ranucci and colleagues in their manuscript entitled “Plasma-Free Strategy For Cardiac Surgery With Cardiopulmonary Bypass In Infants < 10 Kg” studied the role of FFP-free priming for cardiopulmonary by-pass in infants undergoing cardiac operations.

This small propensity matched analysis showed that while avoiding FFP is possible, it results in more pronounced coagulopathy requiring infusions of fibrinogen and PCC. The manuscript is well written and touches an important topic. The conclusions are fair and supported by the data.

Major:

  1. As with many studies on bleeding in cardiac surgery the main issue is the choice of the endpoint. Please provide the rationale for choosing a numerical endpoint of lost blood volume instead of clinical endpoints, such as transfusions. Please provide rationale for choosing the blood loss in 24h instead of 12h, as used in the Universal Definition of Perioperative Bleeding in Cardiac Surgery for adults.

We would like to thank the reviewer for pointing this out. Actually, we decided to choose the chest drain blood volume as the primary endpoint because it represents, especially in pediatric population, a more objective way to evaluate bleeding. Transfusions could be given due to indications other than bleeding and thus bias the analysis of the efficacy of FFP-free versus FFP-based strategies on the hemostasis management.

The 24-hour time point for the postoperative blood loss was chosen instead of the 12-hour (as we usually do for the adults, according to the Universal Definition of Perioperative Bleeding in Cardiac Surgery) because it is the most used in the literature for this kind of patients. The 12-hours and the 48-hours bleeding information was also collected (Table 3).

  1. The study involved small children unable to consent. Did the parents of studied children provide a written informed consent for study participation? Since the research involves minors, please outline the details of informed consent s process.

Parents of the pediatric patients (one or both of them) consent to all the procedures related to cardiac surgery and post-surgery evaluations by dedicated informed consent forms. The possibility to use patients’ clinical data for scientific purposes in anonymous form is there mentioned. Specifically, for this study, the Ethical Committee allowed us to use anonymized patients’ data and prompted to obtain the specific consent whenever possible (follow-up visits, new surgeries or procedures, etc.).

  1. How many patients in each group were operated in deep hypothermic circulatory arrest and how many required open chest management? Can you provide some details on the operations performed?

We would like to thank the reviewer for bringing to the attention the fact that details on the performed surgeries were partial (some technical data about CPB duration, nadir hematocrit and temperature on CPB, RACHS II, the percent of patients undergoing redo and the urgent interventions are present in Table 1). We have now added a paragraph at the beginning of the Results section with a summary of the operations performed.

Answering the specific question of the reviewer, 6 patients of the study underwent deep hypothermic circulatory arrest – 4 in the FFP-free and 2 in the FFP-based group. Unfortunately, the information about the open chest management after surgery is not available among the collected data. This is a good point that we will keep into account in the future studies on pediatric patients.

  1. What was the c-statistics value for the final model.

Is the reviewer referring to the c-statistics of the logistic regression used for propensity matching? If this is the case, after matching the c-statistics should be unable to predict the allocation group. Actually, the value is 0.64 (95% confidence interval 0.49-0.83), therefore confirming the correct propensity matching analysis. However, the usual standard for confirming the correct propensity matching is the assessment of the absolute standardized mean difference that we have shown in Table 1.

  1. Which matching method was used?

As stated in the methodology, we have applied a 1.5:1 based propensity matching, meaning that out of the larger FFP-based group we could extract 27 patients matched with the 18 FFP-free patients. The process of selection followed the usual standard of practice (identification of factors being associated with FFP-based or FFP-free group, assessment of their weight through a logistic regression analysis; calculation of the propensity score and selection of patients with the same or very similar propensity score)

Minor

  1. There is a redundant dot at the end of the abstract.

Redundant dot removed, thank you for the observation.

  1. Page 2, line 56: based on…? Line 59 – experimenting with?

Line 56 has been corrected into “based on”, line 59 – rephrased using “employing” instead of “experiencing”, thank you for pointing out.

Reviewer 3 Report

A very good manuscript. Good flow of thoughts supplemented by neat figures. Perhaps can summarise of the important findings of this study in the conclusion section and what further research can be done in this area of study.

Author Response

We would like to thank the reviewer for his analysis of our manuscript. The main findings of the study are summarized in the first paragraph of the Discussion section and then recalled in the last paragraph, commenting on their clinical implications. A sentence at the end of the manuscript, pointing out the need for further research in order to elucidate the optimal interventions for hemostasis management in pediatric patients undergoing cardiac surgery, has now been added.

Reviewer 4 Report

I read the manuscript "Plasma Free-Free For Cardiac Surgery with Cardiopulmonary Bypass In Infant < 10kg " .

Since the availability of a viscoelastic test was one of the main inclusion criteria suggest to include this in the methodology of the abstract. The title of the manuscript can probably be improved to reflect the study better

Author Response

We would like to thank the reviewer for pointing this out. The abstract has been modified based on his suggestion. The title has been modified specifying the retrospective nature of the study in order to better reflect the topic of the paper.